

# A study of the interactive mediating effect of ADHD and NSSI caused by co-disease mechanisms in males and females

Fang Cheng[1,2,3], Linwei Shi[4], Huabing Xie[5], Beini Wang[2,3], Changzhou Hu[2,3], Wenwu Zhang[2,3], Zhenyu Hu[2,3], Haihang Yu[2,3] and Yiming Wang[1,6]

[1] The Second Affiliated Hospital of Soochow University, Suzhou, China
[2] Department of Psychiatry, Affiliated Kangning Hospital of Ningbo University, Ningbo, China
[3] Department of Psychiatry, Ningbo Kangning Hospital, Ningbo, China
[4] Fuzhou University, Fuzhou, China
[5] People's Hospital of Wuhan University, Wuhan, China
[6] Department of Psychiatry, Affiliated Hospital to Guizhou Medical University, Guizhou, China

Corresponding authors
Haihang Yu,
yuhaihang0414@sina.com
Yiming Wang, 754603457@qq.com

## ABSTRACT

**Background**. Non-suicidal self-injury (NSSI), of which the predisposing factors are complex and diverse, profoundly affects the physical and mental health of young people. Therefore, this work established an NSSI intermediary network model considering the interaction of multiple factors. A mediating effect between attention-deficit/hyperactivity disorder (ADHD) and NSSI, considering the influence of comorbidities, such as depression, anxiety, and impulsive personality, was proposed based on sex differences.

**Methods**. A total of 2,689 middle school students in Ningbo City, Zhejiang Province, China, were randomly sampled and participated in this study. Data regarding their demographic characteristics, attention deficit, hyperactivity/impulsivity, NSSI, anxiety, depression, internet addiction, and other comorbid symptoms were collected and analyzed. After initially screening the data, variables were assessed for significance using a single-factor inter-group difference analytic method, and a binary logistic regression analysis was performed. The intermediary effect of factors influencing NSSI in males and females was also analyzed.

**Results**. The overall NSSI rate was 15.16%. The results showed that the impact of individual impulsivity characteristics (impulsiveness, the ADHD with hyperactivity/impulsivity subtype) on NSSI behavior was not significant (regression results, $P > 0.05$). The degree of association between ADHD with attention deficit and ADHD with comprehension deficit subtypes, and other comorbid symptoms (depression, anxiety, and internet addiction disorder) and NSSI, with odds ratios (ORs) of 7.6/6.42/436.68/3.82/1.86, and 95% bootstrap confidence intervals (CIs) of 4.64, 12.87/3.46, 12.67/137.42, 2659.13/2.32, 6.37/1.31, 2.82, respectively. The results also showed significant effects of ADHD subtypes on comorbid symptoms and the path effects of NSSI ($P < 0.01$). Among them, the mediating effect was the strongest when anxiety was the mediating variable, and the mediating effect of girls was higher than that of boys.

**Conclusion**. The results of this work demonstrated the influence of ADHD symptoms on NSSI behavior. Among patients with ADHD, patients with subtypes with obvious attention deficit characteristics were more likely to exhibit NSSI behavior, whereas the hyperactive impulse subtype had no direct impact on NSSI. We conclude that adolescent impulsivity may not be directly related to NSSI behavior and that impulsive characteristics jointly affect NSSI behavior through a series of NSSI comorbid symptoms. Notably, the probability of symptom onset and the degree of comorbidity was significantly higher in girls than in boys of the same age, and girls were more prone to NSSI behavior. These findings provide effective theoretical support for the prevention and treatment of adolescent NSSI behavior.

## INTRODUCTION

Non-suicidal self-injury (NSSI) refers to the behaviors of intentionally and directly damaging body tissue in the form of skin cuts, scratches, burns, bites, and bruises without suicidal intention (*Klonsky, Victor & Saffer, 2014*). *Perini et al. (2019)* believe that brain regions responsible for common negative emotions can manage negative emotions caused by pain. Therefore, along with the generation and elimination of physical pain, the sense of relief from pain often makes people use self-injury to vent on a psychological level. At present, NSSI usually occurs in youths between the ages of 12 and 15 and reaches a peak in mid-adolescence (*Plener et al., 2015*; *Cipriano, Cella & Cotrufo, 2017*). The complex formation characteristics of NSSI also have prominent regional differences. Data shows that the incidence rate of NSSI among adolescents globally is as high as 17.2% (*Swannell et al., 2014*). However, the incidence of NSSI among adolescents in developed countries such as Scotland, the United States, and Germany is only 13.8%, 15.3%, and 3.1%, respectively, compared with an incidence rate of 22.4% among adolescents in China (*Lang & Yao, 2018*). In addition, evidence (*Nitkowski & Petermann, 2010*; *Turner et al., 2015*; *Buelens et al., 2020a*) shows that the phenomenon of NSSI is related to anxiety, depression, borderline personality disorder (BPD), and other mental disorders. Research has also shown how NSSI can occur with any mental disorder, although anxiety and mood disorders, post-traumatic stress disorder, substance use disorder, eating disorders, and personality disorder comorbidity rates are particularly high (*Cipriano, Cella & Cotrufo, 2017*; *Mohl, 2019*; *Buelens et al., 2020b*). More scholars have also turned their attention to researching sex differences in NSSI (*Bresin & Schoenleber, 2015*; *Victor et al., 2018*). In today's era of increasingly important global public health issues, NSSI behavior has gradually become one of the important reference standards for the physical and mental health of adolescents globally and has increasingly attracted the attention of international scholars.

Attention deficit and hyperactivity disorder (ADHD) refers to a group of syndromes characterized by difficulty with concentration, short duration of attention, hyperactivity, or impulsivity in childhood (*Sonuga-Barke et al., 2013*). Until recently, many people

believed that ADHD was a condition that only appeared in childhood and subsided in adolescence and early adulthood, with little or no lasting impact on adult life (*Hill & Schoener, 1996*). However, increasing evidence (*Wood et al., 1976*) shows that ADHD can continue into adolescence and even adulthood and is accompanied by a series of clinical and psychosocial disorders. The Diagnostic and Statistical Manual (DSM-IV) ADHD standard defined by the American Psychiatric Association is the most widely used standard and describes three subtypes of ADHD according to the main symptom patterns: attention deficit type, hyperactive impulsive type, and combination type (*Battle, 2013*). A study by the World Health Organization's Mental Health Survey (*Lara et al., 2009*) found that the predictive factors of adult ADHD included the subtype of ADHD syndrome in childhood, the severity of symptoms, the existence of comorbid depression, a high incidence of other comorbidities, social adversity, and parental psychopathology. *Biederman et al. (1996)* reported that a family history of ADHD, psychosocial adversity, and co-existing behavioral, emotional, and anxiety disorders were factors predictive of persistence. ADHD often occurs in childhood (*Taylor et al., 1996*), and although the level of ADHD symptoms seems to be lower in adolescence than in childhood, it presents with various complications, such as anxiety, depression, and BPD in adolescence and adulthood, and has a series of complex links with NSSI. For example, *Evans et al. (2022)* found that there may be a correlation between ADHD symptoms and NSSI in veterans. *Brown et al. (2022)* found that the incidence of suicide and NSSI among college students with ADHD was significantly higher than that of college students without ADHD. Their conclusions identified specific risk predictors for students with ADHD.

With the deepening of research, scholars have found that NSSI is not a single independent pathological symptom but has a complex and dynamic pathogenesis involving genetic, biological, mental, psychological, physiological, social, cultural, and other cross-effects (*Kaess et al., 2021*; *Richmond-Rakerd et al., 2019*). Research showed that factors contributing to NSSI can be divided into internal functions, including emotional regulation, ideological regulation, and self-punishment, and interpersonal functions, which mainly convey pain, social influence, and punishment (*Taylor et al., 2018*). However, adolescence is a stage of gradual psychological maturity in which values and judgments are not fully developed, and psychological problems can easily develop. If an adolescent cannot regulate emotions well or cope with stress, depression, and anxiety, other emotional disorders may develop, resulting in self-injurious behavior (*Stewart, Baiden & Theall-Honey, 2014*; *Bentley et al., 2015*; *Ose, Tveit & Mehlum, 2021*). In particular, due to the multifaceted influence of physiological, psychological, and social factors, the influence of ADHD on NSSI appears to differ according to sex. By describing the lifetime risk of NSSI, suicidal ideation (SI), and suicide attempts (SA), *Meza, Owens & Hinshaw (2021)* compared female participants with ($n = 140$) and without ($n = 88$) ADHD and found that ADHD was associated with self-harm during adolescence and early adulthood, especially in females. *Meza, Owens & Hinshaw (2016a)* conducted an in-depth discussion of the longitudinal association of response inhibition, peer preference, and self-harm in young adult women with the reference variable ADHD. *Swanson, Owens & Hinshaw (2014)* conducted a pathway study of self-injurious behavior in young women with ADHD. The study found that

in adolescence, objectively measured impulsivity plus comorbid externalizing symptoms were concurrent, in part mediating the link between childhood ADHD and young adult NSSI. Symptoms internalized in adolescents appeared to partly mediate the relationship between ADHD in children and suicide attempts in young adults, and female patients with ADHD, especially with concurrent childhood impulsivity and persistent symptoms, carried a high risk of self-harm. Psychotic comorbidities and response inhibition were also important mediators of this clinically important longitudinal association. *Balazs et al. (2018a)* similarly found some sex effects in their study of the effects of ADHD and non-suicidal self-harm in a clinical sample of adolescents. The relationship between ADHD symptoms and current NSSI prevalence was entirely mediated by comorbid symptoms in both sexes. The important mediating factors were affective and psychotic symptoms and suicidal tendencies in both sexes, as well as symptoms of alcoholism/dependence disorder in girls.

In summary, there is no clear conclusion regarding whether impulsive characteristics have a direct effect on NSSI, and the symptoms of each comorbid condition (depression, anxiety, and internet addiction) are significant risk factors for NSSI. Although many studies have examined the association between ADHD and NSSI behavior, it is not possible to further clarify the effect of ADHD on NSSI because symptoms in ADHD describe a combination of attention deficit and hyperactive/impulsive behavior characteristics. Thus, to further explore the association between ADHD symptoms and comorbid symptoms and the occurrence of NSSI behavior, this work analyzed 2,689 cross-sectional data, established a gradual and progressive data model, and analyzed the results of each survey. The mechanism of action and contributing effect of the ADHD subtype on NSSI were deeply explored, effectively revealing a complex interaction between ADHD and NSSI and the significant role of sex differences. The results provide strong theoretical guidance for the effective prevention of and response to NSSI.

## METHODS

### Ethics statement

Informed consent was obtained from all participants and their guardians prior to the study's initiation. This included comprehensive information about the study's nature, the potential sensitivity of discussed topics, and strict confidentiality protocols, along with the mandatory reporting requirements. The study protocol was approved by the Ethics Committee of Ningbo Kangning Hospital (approval number: NBKNYY-2020-LC-52), ensuring compliance with the Declaration of Helsinki and relevant ethical guidelines.

### Participants

This cross-sectional study was conducted from September to October 2021 in four districts of Ningbo City, Zhejiang Province, China (Haishu, Jiangbei, Gaoxin, and Yinzhou). We employed a cluster randomized sampling strategy to select a representative sample from regional junior and senior high schools. Three junior high schools and three senior high schools were randomly chosen, ensuring a diverse demographic representation. Selected students from grades 1 to 3 in junior high schools and grades 1 to 2 in senior high schools

were enrolled to participate in the survey. A randomly chosen class from each grade participated.

## Survey instruments and measures

The questionnaire encompassed several domains:

**Basic characteristics**: *Fang & Li (2019)* used a basic characteristics questionnaire to effectively reflect the basic status of participants when studying NSSI behavior and risk factors in patients with depressive disorders. Therefore, in this work, the basic status questionnaire mainly included: age (11–13, 14–16), sex (male, female), physical condition (PC: weak, normal, good), academic performance (AP: poor, average, good, excellent), peer relationships (PR: poor, average, close), and family relationships (FR: separation/divorce, domestic violence, quarrel conflicts, harmony).

**Attention deficit, hyperactivity/impulsivity investigation**: The Chinese version of the Attention Deficit Hyperactivity Disorder SNAP-IV Rating Scale-Parent Version was selected to investigate the characteristics of attention deficit and hyperactivity/impulsive behaviors in middle school students. The scale consists of three subscales: items 1 to 9 are the attention deficit subscale, items 10 to 18 are the hyperactivity/impulsivity subscale, and items 19 to 26 are the oppositional defiance subscale. Each item is answered using a 0–3 scale where: 0 = not present at all; 1 = slightly present; 2 = moderately present, and 3 = extremely present. *Jinbo, Lanting & Ying (2013)* analyzed the reliability and validity testing of the scale in 2013, and the sensitivity of SNAP-IV for diagnosing ADHD was 0.87, and the specificity was 0.79. In this study, only the attention deficit subscale and hyperactivity/impulse subscale were used for evaluation. If the score of any item was ≥2, the participant was judged to have symptoms of the item. If six or more items in the subscale met the criteria, the corresponding ADHD subtype could be diagnosed. This work used ADHD_ HI, ADHD_ AD, and ADHD_ C to represent the hyperactivity/impulsivity subscale, the attention deficit subscale, and the overall scale, respectively.

The Chinese version of the Barratt Impulsiveness Scale (Barratt Impulsiveness Scale-Chinese version 11, BIS-11) (*Li et al., 2011*) was used to observe and analyze the subject's impulsivity characteristics further. The scale has a total of 30 items, which are divided into three subscales: cognitive impulsivity, action impulsivity, and poor planning. For this work, C_Impulsiveness, S_ Impulsiveness and P_ Impulsiveness represent the quantized value of each scale. Each item is scored on a 1–4 scale: (1: never or almost never; 2: occasionally; 3: often; 4: always or almost always).

**Non-suicidal self-injurious behavior:** Data on non-suicidal self-injurious behavior (NSSI) were gathered in line with the methodology previously established by *Cheng et al. (2022)*. This approach specifically involved identifying NSSI based on DSM-5 criteria through a series of true-or-false questions. These questions assessed whether individuals engaged in more than five acts of intentional self-injury in the past year, which were not life-threatening but resulted in minor to moderate physical harm like bleeding, bruising, or pain. Additionally, it was determined whether these actions were aimed at relieving psychological stress, resolving interpersonal issues, or eliciting a positive emotional response. An affirmative answer to these questions was indicative of NSSI behavior.
**Investigation of comorbid symptoms of anxiety, depression, and internet addiction:** BPD is often accompanied by common conditions, such as depressive disorder (DD), anxiety neurosis (AN), and internet addiction disorder (IAD). However, there has been limited in-depth discussion regarding the effects of these conditions on attention deficit, hyperactivity, and NSSI. Therefore, the Screen for Child Anxiety-related Emotional Disorders (SCARED) (*Wang et al., 2002*), compiled by Birmaher, was chosen for the self-assessment of anxiety in children and adolescents aged 8–16 years. The scale has a total of 41 items covering five factors (generalized anxiety, somatization/panic, dissociative anxiety, social phobia, and school phobia), each graded on three levels, where 0 means no symptoms, 1 means partial anxiety, and 2 means frequent anxiety. A total score >23 indicates that the subject has anxiety.

The Depression Self-Rating Scale for Children (DSRSC), designed by Birleson (*Fang & Li, 2019*), was used to evaluate symptoms of depression in middle school students. The scale consists of 18 items. Each item is scored using a 0–2 scale where 0 = no, 1 = sometimes, and 2 = often. Total scores of less than 15 points indicate no symptoms of depression, total scores between 16–20 points indicate there may be depressive symptoms, and scores >20 indicate the presence of depressive symptoms.

Finally, the Internet Addiction Improvement Index (IAII) (*Zhang et al., 2021*) was used to evaluate the tendency and severity of the subject's internet addiction. The scale has a total of 20 items and adopts a five-level scoring method where 1 is hardly, 2 is occasionally, 3 is sometimes, 4 is often, and 5 is always. Higher scores indicate more serious internet addiction. IAD is determined if the score is $\geq$ 50.

## Potential bias

Our study identified several potential sources of bias. (1) Selection bias: A risk of selection bias might exist due to the cluster randomized sampling method used. This would occur if the schools and students selected are not fully representative of the broader population. (2) Response bias: A risk of response bias might exist given the self-reported nature of the survey where participants might provide socially desirable answers or misreport their behaviors and experiences. (3) Information bias: The accuracy of the collected data might be affected by the way the questions are understood and interpreted by the respondents.

We implemented several strategies to address these potential biases: (1) Minimizing selection bias: We used a random selection process for the schools and classes to ensure a representative sample of the target population. Additionally, we included a diverse range of schools from different districts to enhance representativeness. (2) Reducing response bias: We assured participants of strict confidentiality and anonymity to mitigate response bias. This was intended to encourage honest and accurate responses. The survey was administered in a controlled environment to minimize external influences. (3) Limiting information bias: The questionnaire was carefully designed to be clear and understandable. A pilot test was conducted prior to the main study to identify and rectify any ambiguous questions. Trained psychiatrists explained the survey requirements to participants, ensuring that the questions were comprehended correctly. (4) Data quality control: A pre-test was administered to participants before the survey. After completing the questionnaire, the

doctors of the main test collected them, entered and screened two questionnaires, and then checked the questionnaire to ensure the quality of the answers.

## Sample size

The study's sample size was determined based on established international questionnaire design principles (*Xie et al., 2023*). These guidelines suggest that the sample size should be approximately five to 20 times the number of items in the questionnaire to ensure adequate data representation and reliability. Anticipating the possibility of non-responses or invalid questionnaires, which typically account for about 10% of the total surveys distributed, we adjusted our target sample size upward by 10%. Therefore, the final sample size was set to be 5.5 times the number of questionnaire items, ensuring a sufficiently large sample to maintain the robustness of our study's findings, even with the anticipated non-response or invalidity rate. Given that our study was comprised of 142 items, the minimum sample size was calculated to be 781 (142 items × 5.5). Finally, a total of 2,784 questionnaires were administered, with 2,753 questionnaires returned, yielding a response rate of 98.89%. Of these questionnaires, 64 had more than 30% missing data and logical errors and were thus recorded as invalid. There were 2,689 valid questionnaires, resulting in an effective completion rate of 96.59%.

## Data analysis methods

After logic checking and proofreading, we used R−4.2.1 (open source programming language) and R studio for Windows (open source IDE) to process and analyze the data. First, we conducted a series of chi-squared tests and rank-sum tests to evaluate inter-group differences between the relevant classification variables and continuous variables and NSSI diagnostic status and sex. We also comprehensively described and characterized the distributions of the various types of data. Values are presented as means ± standard deviations for overall scores, medians and interquartile ranges for grouping data for continuous variables, and numbers (%) for categorical variables.

Second, to assess risk factors for NSSI, we conducted a series of binary logistic regressions to assess whether variables such as attention deficit, hyperactivity/impulsive behavior characteristics, comorbid symptoms (anxiety, depression, and internet addiction) and subjects' demographic characteristics were predictive of NSSI and calculated odds ratios (ORs) for each predictor variable. At the same time, we used a stepwise regression method, using Akaike information criterion (AIC) (*Akaike, 1974*) as the evaluation method, to select the minimum AIC using forward and backward stepwise regression methods to delete any collinear variables in the model and obtain the binary prediction model of each NSSI variable.

Finally, based on the variables included in the obtained binary prediction model, we explored whether treatment of ADHD could change the comorbid symptoms of patients, thus changing the phenomenon of NSSI. We also explored the common mechanisms of ADHD symptoms and comorbid symptoms affecting NSSI and tested the path through the Sobel test (*Preacher & Hayes, 2008*) to preliminarily examine intermediary interactions between various variables.

This work used 95% confidence intervals to describe the statistical results, and *p*-values of less than 0.05 were considered statistically significant.

## RESULTS

### Participant characteristics and prevalence of NSSI
A total of 2,689 school students participated in this study, of whom 435 (15.16%) had NSSI behavior. The number of participants by age group included 817 in the 11–13-year-old group (97 in the NSSI group, 11.87%) and 1872 in the 14–16-year-old group (338 in the NSSI group, 18.06%). The sex characteristics of the sample were 1,157 males (78 in the NSSI group, 6.74%) and 1,532 females (357 in the NSSI group, 23.30%). The age and sex of the middle school students were different from those of primary school students ($P < 0.05$).

### Descriptive and single-factor intergroup analysis
Table 1 presents the distribution of survey variables among secondary school students by sex and NSSI behavior. Significant differences ($P < 0.05$) were observed in most variables between the NSSI and non-NSSI groups. However, no significant difference was found between males and females in S_impulsiveness ($P = 0.315$). Students with NSSI behavior had poorer peer and parent relationships and higher anxiety, depression, and internet addiction scores. ADHD_C and ADHD_AD symptoms were more prevalent in the NSSI group, whereas ADHD_HI symptoms were less common. No significant differences were observed in the BIS assessment of each impulsiveness component.

In terms of sex, no significant differences were noted between males and females in basic characteristics, IAD, and BIS scores. However, girls showed significantly higher scores and prevalence rates of anxiety, depression, and ADHD symptoms compared to boys.

### Predictor binary logistic regression analysis
Binary logistic regression analysis was conducted to assess the effect of independent predictors on dichotomized NSSI (Table 2). Basic attributes, including age, physical condition, academic performance, peer relationships, and family relationships, did not show significant effects on NSSI ($P > 0.05$). Additionally, impulsivity (S_impulsiveness, C_impulsiveness, P_impulsiveness) and ADHD symptom subtype-hyperactive impulsive type had no significant effect on NSSI ($P > 0.05$). The overall AIC score of the model was 824.6.

Stepwise regression analysis was used to refine the model and identify the strongest predictors of NSSI (Table 3). Depression, anxiety disorder, and a positive diagnosis of internet addiction were significantly positively correlated with NSSI behaviors ($P < 0.001$). The OR for each symptom positively associated with NSSI was 436.68 for depression, 3.82 for anxiety disorder, and 1.86 for internet addiction, with 95% boot CIs of 137.42, 2659.13/2.32, and 6.37/ 1.31, 2.82. For ADHD symptoms, ADHD_AD and ADHD_HI positivity were significantly correlated with NSSI ($P < 0.001$), with ORs of 7.6 and 6.42, respectively. Sex also influenced NSSI occurrence, with being male sex negatively associated compared to being female ($P = 0.077$), with an OR of 0.7 and 95% boot CI of 0.47, 1.04. The model's final AIC score was 807.51.

**Table 1  Characteristics of various variables among students according to NSSI status and gender.**

|  | Total | Non-NSSI | NSSI | *P*-value | Female | Male | *P*-value |
|---|---|---|---|---|---|---|---|
| Total | 2,689 | 2,254 | 435 |  | 1,532 | 1,157 |  |
| AGE |  |  |  | <0.001 |  |  | <0.001 |
| 14–16 | 1,872 (69.6) | 1,534 (68.1) | 338 (77.7) |  | 1,108 (72.3) | 764 (66) |  |
| 11–13 | 817 (30.4) | 720 (31.9) | 97 (22.3) |  | 424 (27.7) | 393 (34) |  |
| AP |  |  |  | <0.001 |  |  | 0.004 |
| excellent | 254 (9.4) | 201 (8.9) | 53 (12.2) |  | 156 (10.2) | 98 (8.5) |  |
| good | 1,151 (42.8) | 1,018 (45.2) | 133 (30.6) |  | 634 (41.4) | 517 (44.7) |  |
| average | 998 (37.1) | 831 (36.9) | 167 (38.4) |  | 554 (36.2) | 444 (38.4) |  |
| poor | 286 (10.6) | 204 (9.1) | 82 (18.9) |  | 188 (12.3) | 98 (8.5) |  |
| PR |  |  |  | <0.001 |  |  | <0.001 |
| close | 1,207 (44.9) | 1,067 (47.3) | 140 (32.2) |  | 663 (43.3) | 544 (47) |  |
| average | 1,420 (52.8) | 1,164 (51.6) | 256 (58.9) |  | 816 (53.3) | 604 (52.2) |  |
| poor | 62 (2.3) | 23 (1) | 39 (9) |  | 53 (3.5) | 9 (0.8) |  |
| FR |  |  |  | <0.001 |  |  | <0.001 |
| harmony | 2,060 (76.6) | 1,840 (81.6) | 220 (50.6) |  | 1,119 (73) | 941 (81.3) |  |
| quarrel conflict | 386 (14.4) | 261 (11.6) | 125 (28.7) |  | 248 (16.2) | 138 (11.9) |  |
| domestic violence | 41 (1.5) | 19 (0.8) | 22 (5.1) |  | 31 (2) | 10 (0.9) |  |
| separation/divorce | 202 (7.5) | 134 (5.9) | 68 (15.6) |  | 134 (8.7) | 68 (5.9) |  |
| PC |  |  |  | <0.001 |  |  | <0.001 |
| normal | 2,073 (77.1) | 1,803 (80) | 270 (62.1) |  | 1,142 (74.5) | 931 (80.5) |  |
| good | 553 (20.6) | 405 (18) | 148 (34) |  | 341 (22.3) | 212 (18.3) |  |
| weak | 63 (2.3) | 46 (2) | 17 (3.9) |  | 49 (3.2) | 14 (1.2) |  |
| Anxiety (CV) | 29.7 (20.7) | 22 (13,32) | 68 (52,76) | <0.001 | 30 (18,49) | 20 (12,30) | <0.001 |
| Anxiety (DV) |  |  |  | <0.001 |  |  | <0.001 |
| Positive | 1,322 (49.2) | 911 (40.4) | 411 (94.5) |  | 909 (59.3) | 413 (35.7) |  |
| Negative | 1,367 (50.8) | 1,343 (59.6) | 24 (5.5) |  | 623 (40.7) | 744 (64.3) |  |
| Depression (CV) | 12 (6.8) | 10 (6,14) | 22 (19,25) | <0.001 | 13 (8,18) | 9 (6,14) | <0.001 |
| Depression (DV) |  |  |  | <0.001 |  |  | <0.001 |
| Positive | 803 (29.9) | 371 (16.5) | 432 (99.3) |  | 581 (37.9) | 222 (19.2) |  |
| Negative | 1,886 (70.1) | 1,883 (83.5) | 3 (0.7) |  | 951 (62.1) | 935 (80.8) |  |
| IAD(CV) | 29.4 (21.9) | 22 (10,37) | 57 (40,70) | <0.001 | 28 (11.8,46.2) | 24 (12,39) | <0.001 |
| IAD (DV) |  |  |  | <0.001 |  |  | <0.001 |
| Positive | 418 (16) | 198 (8.9) | 220 (57.1) |  | 281 (19) | 137 (12) |  |
| Negative | 2,202 (84) | 2,037 (91.1) | 165 (42.9) |  | 1,195 (81) | 1,007 (88) |  |
| ADHD_C | 21.3 (16.2) | 15 (9,24) | 49 (22,58) | <0.001 | 18 (10,34) | 16 (10,24) | <0.001 |
| ADHD_AD | 9.4 (6.5) | 7 (4,11) | 20 (10,24) | <0.001 | 9 (5,14) | 7.1 (4,11) | <0.001 |
| ADHD_HI | 5.4 (5.5) | 3 (1,6.1) | 12 (6,16) | <0.001 | 4 (1,10) | 3 (1,7) | <0.001 |
| ADHD (DV) |  |  |  | <0.001 |  |  | <0.001 |
| Hybrid | 128 (4.8) | 40 (1.8) | 88 (20.2) |  | 100 (6.5) | 28 (2.4) |  |
| Hyperactivity | 28 (1) | 24 (1.1) | 4 (0.9) |  | 16 (1) | 12 (1) |  |
| Attention Defects | 299 (11.1) | 122 (5.4) | 177 (40.7) |  | 232 (15.1) | 67 (5.8) |  |

**Table 1** (*continued*)

|  | Total | Non-NSSI | NSSI | *P*-value | Female | Male | *P*-value |
|---|---|---|---|---|---|---|---|
| Normal | 2,234 (83.1) | 2,068 (91.7) | 166 (38.2) |  | 1,184 (77.3) | 1,050 (90.8) |  |
| S_ Impulsiveness | 23.3 (2.4) | 23 (22,25) | 23 (21,25) | 0.007 | 23 (22,25) | 23 (22,25) | 0.315 |
| C_ Impulsiveness | 21.4 (5.2) | 21 (18,25) | 19 (16,23) | <0.001 | 21 (18,24) | 21 (18,25) | <0.001 |
| P_ Impulsiveness | 25.7 (2.9) | 26 (24,28) | 26 (24,28) | 0.04 | 26 (24,28) | 26 (24,28) | 0.001 |

**Notes.**

mean (SD) for global data; median (IQR) for subgroup data; number (%) for categorical variables.

CV, continuous variables; DV, discontinuous variables.

**Table 2 Results of the logistic regression: effect of all survey variables on NSSI ($n = 2,689$).**

| Variables | Estimate | Std.ERROR | Z value | P value | OR |
|---|---|---|---|---|---|
| Gender | −0.36 | 0.20 | −1.77 | 0.077 | 0.7(0.47~1.04) |
| ADHD attention defects | 1.92 | 0.29 | 6.70 | 0.000 | 6.81(3.94~12.14) |
| ADHD hyperactivity | 0.52 | 0.79 | 0.66 | 0.512 | 1.68(0.35~8.73) |
| ADHD hybrid | 1.72 | 0.37 | 4.62 | 0.000 | 5.61(2.76~12.02) |
| AGE 11-13 | −0.05 | 0.21 | −0.22 | 0.823 | 0.95(0.63~1.44) |
| PC good | 0.15 | 0.23 | 0.66 | 0.510 | 1.16(0.74~1.82) |
| PC weak | −0.72 | 0.59 | −1.21 | 0.226 | 0.49(0.16~1.66) |
| AP good | 0.10 | 0.31 | 0.32 | 0.747 | 1.11(0.6~2.04) |
| AP average | 0.30 | 0.31 | 0.96 | 0.336 | 1.35(0.74~2.47) |
| AP poor | 0.58 | 0.39 | 1.49 | 0.136 | 1.78(0.84~3.82) |
| PR average | 0.03 | 0.19 | 0.14 | 0.891 | 1.03(0.71~1.49) |
| PR poor | 0.59 | 0.63 | 0.93 | 0.354 | 1.8(0.56~6.95) |
| FR quarrel conflict | 0.05 | 0.26 | 0.20 | 0.842 | 1.05(0.63~1.75) |
| FR domestic violence | 0.53 | 0.74 | 0.72 | 0.474 | 1.69(0.42~7.57) |
| FR separation/divorce | 0.59 | 0.35 | 1.68 | 0.094 | 1.8(0.91~3.64) |
| S_BIS | −0.03 | 0.03 | −0.92 | 0.356 | 0.97(0.9~1.04) |
| C_BIS | 0.00 | 0.02 | 0.09 | 0.929 | 1(0.96~1.05) |
| P_BIS | −0.02 | 0.04 | −0.49 | 0.622 | 0.98(0.92~1.05) |
| DEPRESS positive | 6.04 | 0.72 | 8.41 | 0.000 | 420.86(131.91~2567.36) |
| ANXIOUS positive | 1.34 | 0.26 | 5.21 | 0.000 | 3.82(2.34~6.43) |
| IAD positive | 0.62 | 0.20 | 3.11 | 0.002 | 1.86(1.25~2.74) |
| AIC = 824.6 |  |  |  |  |  |

**Table 3 Results of the logistic regression: effect of selected variables on NSSI behavior ($n = 2,689$).**

| Variables | Estimate | Std.ERROR | Z value | P value | OR |
|---|---|---|---|---|---|
| Gender male | −0.35 | 0.20 | −1.77 | 0.077 | 0.7(0.47~1.04) |
| ADHD attention defects | 2.03 | 0.26 | 7.81 | 0.000 | 7.6(4.64~12.12.87) |
| ADHD hyperactivity | 0.51 | 0.78 | 0.66 | 0.511 | 1.67(0.36~8.54) |
| ADHD hybrid | 1.72 | 0.37 | 4.62 | 0.000 | 5.61(2.76~12.02) |
| DEPRESS positive | 6.08 | 0.72 | 8.48 | 0.000 | 436.68(137.42~2659.13) |
| IAD positive | 0.66 | 0.19 | 3.36 | 0.001 | 1.93(1.31~2.82) |
| AIC = 807.51 |  |  |  |  |  |

## Correlation analysis

In the previous section, univariate analysis and logistic regression analyses were performed to assess the influence of the sample's basic characteristics, anxiety and depression and Internet addiction on NNSI behavior. Preliminary screening eliminated variables such as age, PC, PR, FR, and impulsiveness that were poorly correlated with the NSSI phenomenon. The logistic regression results showed that NSSI comorbid symptoms, including anxiety, depression, internet addiction, and ADHD, played a determinant role in the predictive model of NSSI behaviors, while sex also exerted a degree of influence on the occurrence of NSSI behaviors.

Correlation analysis was used to analyze the degree of association and significance among the variables influenced by NSSI according to sex to further clarify the strength of correlations among the variables, and Pearson's correlation coefficients were used to effectively characterize the strength of correlations among the variables. NSSI, AN, DD, IAD, ADHD_AD, ADHD_HI, and ADHD_C all showed significant positive correlations (all $P < 0.001$). The correlation coefficients for the variables for males and females are shown in Figs. 1A, 1B and Table 4 and show that the degree of association between the variables for females was stronger than for males.

## Mediating effects

A pathway distribution network of NSSI influencing factors based on multiple mediating effects was proposed in this work to further explore the influence of ADHD symptoms on NSSI behavior and clarify the ADHD subtype factors that most influence NSSI behavior. The effect of mediating effects behavior of ADHD-comorbid symptoms-NSSI at the level of sex variability is discussed in depth. Notably, to further quantify the interactions and relationships between the study variables and incorporate the unidirectional nature of the scale scores used in this work, mediating effects were calculated using the scores corresponding to each disease symptom for the analysis.

Combining the logistic regression and correlation analysis results, the mediating effect of ADHD-comorbid symptoms on NSSI was analyzed at the sex level, refining the influence of various ADHD subtypes on NSSI, anxiety, depression, and internet addiction. Schematic diagrams (Figs. 2 and 3) depict the mediation models for males and females, showing significant relationships between ADHD subtypes (ADHD_AD, ADHD_HI, and ADHD_C) and comorbid symptoms in predicting NSSI. Regression coefficients and CIs for these relationships are detailed in the figures.

Table 5 summarizes the potential mediating effects in the model. Indirect effects were tested using bootstrap estimation methods (Olfson et al., 2005). In this model, c represents the total effect of the independent variable X on the dependent variable Y without the mediating variable M, a represents the effect of X on M, b is the effect of M on Y, and ab is the mediating effect. The term c is the direct effect of X on Y when both a and b are significant. A model was partially mediated if ab and c had the same sign and both were significant, as depicted in Figs. 2 and 3.

For both sexes, anxiety, neurosis, depressive disorder, and internet addiction played a partially mediating role in the relationship between various ADHD behaviors and NSSI.
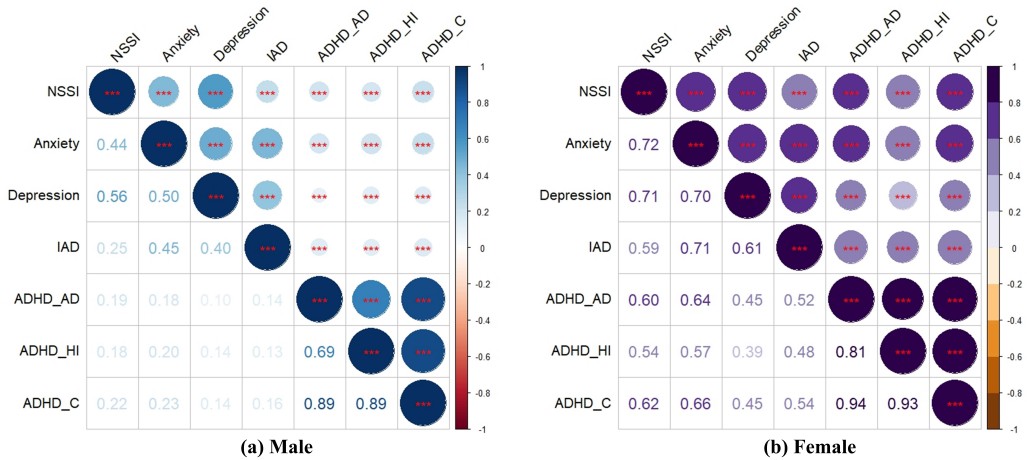

**(a) Male**                                                    **(b) Female**

**Figure 1   The correlation heat map of NSSI impact variables by gender.**

The effectiveness ratios calculated using Eq. (1) indicated the weight ratio of the mediating effect in the model, with 95% bootstrap CIs denoting the weight ratio of the 95% CI from bootstrap sampling. A significant mediating effect was confirmed as the CIs did not include 0.

$$E_f = a*b/c. \tag{1}$$

## Specific mediating effects and sex differences

The direct effects of ADHD subtypes on NSSI decreased after introducing the mediating variables, indicating significant mediation. In males, the intermediary effects and 95% bootstrap CIs for various pathways from ADHD_AD, ADHD_HI, and ADHD_C to NSSI through AN, DD, and IAD showed a decrease in direct effects, confirming the mediation. Similarly, in females, the mediating effects were even more pronounced, with larger decreases in direct effects and significant intermediary effects across all pathways.

In summary, our results indicated that:

(1) The mediating effect of ADHD_HI scores on NSSI behavior through comorbid symptoms was more significant than that of ADHD_AD and ADHD_C.

(2) Among the comorbid symptoms, AN exhibited the most significant mediating effect on each ADHD subtype, followed by DD and IAD.

(3) The degree of mediating effect was consistently higher in female groups compared to male groups.

These findings underscore the complex interplay between ADHD symptoms and comorbid conditions in influencing NSSI behavior, with significant variations observed across sexes. The significant mediating roles of anxiety, depression, and internet addiction highlight the importance of considering these factors in understanding and addressing NSSI behaviors in adolescents with ADHD.

**Table 4  Correlation coefficients of variables impacting NSSI by gender.**

|  | NSSI | AN | DD | IAD | ADHD_AD | ADHD_HI | ADHD_C |
|---|---|---|---|---|---|---|---|
| **Female** | | | | | | | |
| NSSI | 1 | | | | | | |
| AN | 0.72[***] | 1 | | | | | |
| DD | 0.71[***] | 0.70[***] | 1 | | | | |
| IAD | 0.59[***] | 0.71[***] | 0.61[***] | 1 | | | |
| ADHD_AD | 0.60[***] | 0.64[***] | 0.45[***] | 0.52[***] | 1 | | |
| ADHD_HI | 0.54[***] | 0.57[***] | 0.39[***] | 0.48[***] | 0.81[***] | 1 | |
| ADHD_C | 0.62[***] | 0.66[***] | 0.45[***] | 0.54[***] | 0.94[***] | 0.93[***] | 1 |
| **Male** | | | | | | | |
| NSSI | 1 | | | | | | |
| AN | 0.44[***] | 1 | | | | | |
| DD | 0.56[***] | 0.50[***] | 1 | | | | |
| IAD | 0.25[***] | 0.45[***] | 0.40[***] | 1 | | | |
| ADHD_AD | 0.19[***] | 0.18[***] | 0.10[***] | 0.14[***] | 1 | | |
| ADHD_HI | 0.18[***] | 0.20[***] | 0.14[***] | 0.13[***] | 0.69[***] | 1 | |
| ADHD_C | 0.22[***] | 0.23[***] | 0.14[***] | 0.16[***] | 0.89[***] | 0.89[***] | 1 |

**Notes.**
[***] $p < 0.001$.

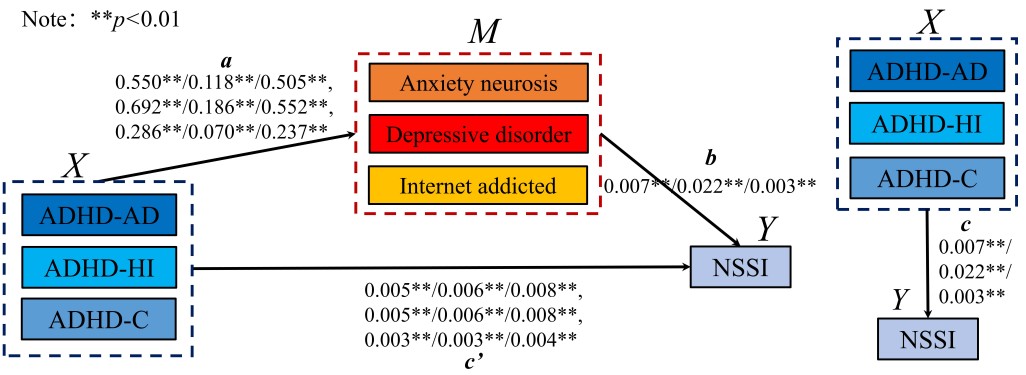

**Figure 2  The mediating model of male.** ** $p < 0.01$.

## DISCUSSION

### Overview of NSSI prevalence and public health implications

The high incidence and harmfulness of NSSI among adolescents have made it a serious public health problem globally. More seriously, NSSI patients and others may experience sustained physical injury (*Preacher & Hayes, 2008*) and face a greater risk of death from suicide (*Asarnow et al., 2011*). In this study, data from 2,689 students (43.03% male and 56.97% female) were collected, of whom 15.16% were positive for NSSI behavior. In existing studies, the detection rate of NSSI in Chinese teenagers was 5.44%–23.2% (*Xu et al., 2020*), consistent with our survey data. At the same time, we found that 357 female students (23.30%) had NSSI behavior, which was significantly higher than the rate for

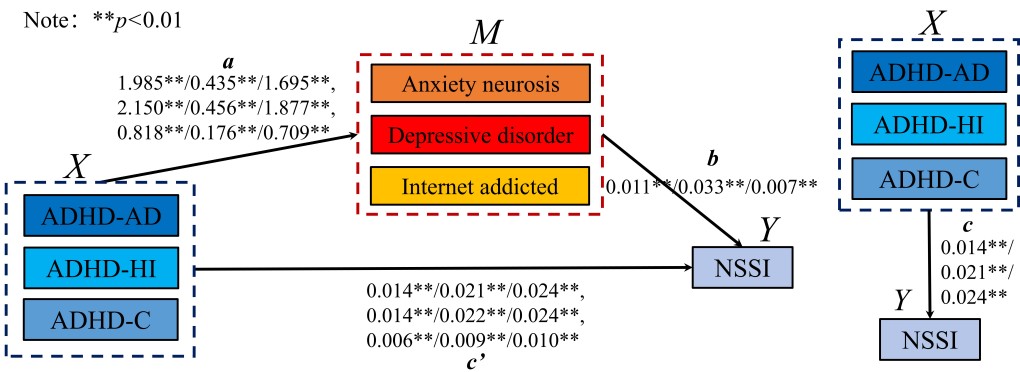

**Figure 3  The mediating model of female.**

**Table 5  Summary of intermediary role tests.**

| Mediating term | c Total effect | a | b | a*b Intermediary effect | c' Direct effect | a*b (95% Boot CI) | Effectiveness ratio (a*b/c) |
|---|---|---|---|---|---|---|---|
| | | | Male | | | | |
| ADHD_AD → AN → NSSI | 0.009** | 0.550** | 0.007** | 0.004 | 0.005** | 0.041–0.116 | 40.610% |
| ADHD_AD → DD → NSSI | 0.009** | 0.118** | 0.022** | 0.003 | 0.006** | 0.018–0.089 | 28.759% |
| ADHD_AD → IAD → NSSI | 0.009** | 0.505** | 0.003** | 0.001 | 0.008** | 0.014–0.048 | 16.259% |
| ADHD_HI → AN → NSSI | 0.010** | 0.692** | 0.007** | 0.005 | 0.005** | 0.049–0.124 | 46.972% |
| ADHD_HI → DD → NSSI | 0.010** | 0.186** | 0.022** | 0.004 | 0.006** | 0.040–0.109 | 41.465% |
| ADHD_HI → IAD → NSSI | 0.010** | 0.552** | 0.003** | 0.002 | 0.008** | 0.013–0.048 | 16.441% |
| ADHD_C → AN → NSSI | 0.004** | 0.286** | 0.007** | 0.002 | 0.003** | 0.058–0.137 | 42.592% |
| ADHD_C → DD → NSSI | 0.004** | 0.070** | 0.022** | 0.002 | 0.003** | 0.041–0.111 | 34.782% |
| ADHD_C → IAD → NSSI | 0.004** | 0.237** | 0.003** | 0.001 | 0.004** | 0.017–0.051 | 15.185% |
| | | | Female | | | | |
| ADHD_AD → AN → NSSI | 0.035** | 1.985** | 0.011** | 0.021 | 0.014** | 0.334–0.396 | 60.349% |
| ADHD_AD → DD → NSSI | 0.035** | 0.435** | 0.033** | 0.014 | 0.021** | 0.223–0.266 | 40.616% |
| ADHD_AD → IAD → NSSI | 0.035** | 1.695** | 0.007** | 0.011 | 0.024** | 0.168–0.222 | 32.312% |
| ADHD_HI → AN → NSSI | 0.038** | 2.150** | 0.012** | 0.025 | 0.014** | 0.319–0.383 | 64.798% |
| ADHD_HI → DD → NSSI | 0.038** | 0.456** | 0.035** | 0.016 | 0.022** | 0.204–0.247 | 41.954% |
| ADHD_HI → IAD → NSSI | 0.038** | 1.877** | 0.008** | 0.014 | 0.024** | 0.176–0.228 | 37.469% |
| ADHD_C → AN → NSSI | 0.015** | 0.818** | 0.010** | 0.009 | 0.006** | 0.335–0.399 | 58.660% |
| ADHD_C → DD → NSSI | 0.015** | 0.176** | 0.032** | 0.006 | 0.009** | 0.222–0.263 | 39.080% |
| ADHD_C → IAD → NSSI | 0.015** | 0.709** | 0.006** | 0.005 | 0.010** | 0.165–0.222 | 30.962% |

**Notes.**
$^{**}p < 0.01$.

male students ($n = 78$; 6.74%). Existing studies have reported inconsistencies regarding sex differences, although some research showed no significant difference in the incidence of NSSI between men and women (*Swannell et al., 2014*; *Klonsky, Oltmanns & Turkheimer, 2003*; *Nock et al., 2006*). However, recent research results showed that the incidence of NSSI in women is significantly higher due to the influence of physiological, psychological,

and social factors (*Barrocas et al., 2012*; *Sornberger et al., 2012*). In a patriarchal society, women's legitimate interests cannot be guaranteed to varying degrees. When demands cannot be reasonably responded to in society or at home, women tend to adopt negative means to numb themselves and even gain attention. At the same time, physiological differences also exacerbate the occurrence of this phenomenon. Therefore, the basis and bias of object selection are crucial in the research and implementation of response measures for NSSI. The research results can better guide us to invest more attention in women's NSSI tendencies.

## NSSI in relation to borderline personality disorder and impulse control

Before being included in the DSM-5, NSSI was considered to be a symptom of BPD (*Battle, 2013*). However, an important feature of BPD is impulse control dysfunction (*Evren et al., 2012*; *Lieb et al., 2004*). Therefore, NSSI has been considered to be related to impulse control problems. As an effective way for individuals to regulate aversion (*Armey, Crowther & Miller, 2011*; *Klonsky, 2007*), people with impulsive personality characteristics may have a high risk of participating in NSSI. However, recent studies provided conflicting results regarding whether impulsivity is a risk factor for NSSI behavior (*Bornovalova et al., 2011*; *Chapman et al., 2009*). The results of this research showed that adolescents' impulsivity may not be directly related to the occurrence of NSSI behavior, as scores on the BIS-11 questionnaire for measures of cognition, action, and impulsivity were not significantly positively correlated with the occurrence of NSSI. Glenn reported a similar finding, but it was not consistent with current research results (*Victor & Klonsky, 2014*). For instance, studies found no significant difference between individuals who engaged in NSSI and those who did not engage in NSSI based on their scores on the multidimensional scale of impulsivity (*Glenn & Klonsky, 2010*; *Janis & Nock, 2009*; *Closkey et al., 2012*). In some studies, when risk factors related to NSSI, such as childhood trauma experience, age, borderline personality characteristics, material dependence, depression, and anxiety psychosis, were controlled for, the relationship between impulsive behavior and NSSI seemed to subside to some extent (*Evren et al., 2012*; *Carli et al., 2010*; *Rodav, Levy & Hamdan, 2014*; *Sacks et al., 2008*). The results of longitudinal studies of the relationship between impulsive behavior and NSSI were inconclusive s (*Glenn & Klonsky, 2011*; *Peterson & Fischer, 2012*). The results of a study on the relationship between impulsivity, alcoholism, drug abuse, and NSSI (*Magid & Colder, 2007*; *Papachristou et al., 2013*; *Coskunpinar, Dir & Cyders, 2013*) showed that the influence of impulsivity on NSSI was often accompanied by other health risk behaviors or presented in other negative ways, which to a certain extent, constituted a significant parameter of NSSI. When other variables affecting the relationship are introduced, the relationship between impulsivity and NSSI becomes mixed. *Glenn & Klonsky (2010)* and *Arens, Gaher & Simons (2012)* found a close relationship between a negative sense of urgency and multiple risk factors related to NSSI (negative emotion, depression, alcoholism, and child abuse) to varying degrees. While these risk factors may be aspects of an impulsive personality, it does not mean that impulsivity is directly related to NSSI. The data in this study did not support a direct correlation between impulse

characteristics and NSSI. It is interesting that the series of behaviors and psychological states triggered by impulsive characteristics seemed to be closely related to NSSI, and this conclusion is also supported to varying degrees by previous research. Therefore, when dealing with NSSI patients, relevant personnel should pay more attention to their underlying psychological states of depression, anxiety, and material dependence, as well as risk factors such as alcohol and drug abuse. Impulsive characteristics often lead to a series of states and behaviors that can directly affect NSSI.

## ADHD and its connection to NSSI

ADHD is a common mental disorder in children and adolescents. Its main symptoms are inattention, hyperactivity, and impulsivity (*Balazs et al., 2018a*), and ADHD often continues to different degrees after adulthood. According to one study (*Faraone, Biederman & Mick, 2006*), approximately two-thirds of patients with ADHD in childhood/adolescence continued to have different degrees of dysfunction in adulthood. Several other studies (*Danckaerts et al., 2010*; *Velo et al., 2013*; *Dallos et al., 2017*) showed that the quality of life of ADHD patients was significantly lower than that of non-ADHD patients. In recent years, researchers have become increasingly interested in the possible link between ADHD and NSSI (*Bentley et al., 2015*; *Meza, Owens & Hinshaw, 2016a*; *Meszaros, Horvath & Balazs, 2017*; *Hinshaw et al., 2012*). *Taylor, Boden & Rucklidge (2014)* found a significant correlation between the severity of ADHD symptoms and NSSI behavior in a normal population, which was modulated by comorbidities, such as affective, anxiety, drug and alcohol abuse disorders, and emotionally focused coping styles. *Meza, Owens & Hinshaw (2016a)* and *Swanson, Owens & Hinshaw (2014)* reported that adolescence and other external disorder symptoms played a significant role in the relationship between ADHD symptoms and the prevalence of NSSI. In studying the relationship between ADHD and NSSI, scholars often take ADHD as a whole to study its impact on NSSI and do not examine ADHD subtypes. At present, three subtypes of ADHD are described in the industry's evaluation system: inattentive, hyperactive and impulsive, and combined (*Battle, 2013*). However, most research has focused on the impact of ADHD overall. In this work, we further refined the causes of the effects of ADHD symptoms on NSSI behavior and found that among patients with ADHD, subtypes with obvious attention deficit characteristics had a significant positive correlation with the occurrence of NSSI behavior, whereas subtypes with hyperactivity did not show a significant correlation with NSSI behavior. Thus, attention deficit characteristics may be the main reason for the greater risk of NSSI behavior. In studying ADHD symptoms, the calm hyperactivity subtype of patients with NSSI behavior is consistent with the above impulse study. It is also worth noting that attention deficit may result in psychological and physiological problems during adolescence (*Klassen, 2014*; *Gnanavel & Robert, 2013*; *Stewart et al., 2018*; *Hafner, Maurer & Wander Heiden, 2013*). This discovery provides further guidance for the correlation research and treatment of ADHD and NSSI. Attention should be paid to the attention deficit subtype of patients as it may further trigger NSSI behavior.

## Mediating role of psychological and physiological comorbidities

In addition, risk factors other than ADHD will interact with ADHD to impact NSSI behavior (*Meza, Owens & Hinshaw, 2016b*; *Balazs et al., 2018b*; *Meszaros et al., 2020*). The role of psychological and physiological comorbidities, such as anxiety neurosis, depressive disorder, and internet addiction, was examined in the process of refining the influence of multiple ADHD subtypes on NSSI to explore the possible mediating effects. In this work, we conducted an in-depth analysis of sex differences and the mediating effect of ADHD comorbid symptoms on NSSI behavior. The results showed that ADHD symptoms could affect the occurrence of NSSI behavior through multiple comorbid symptoms. In further detail, ADHD_HI, representing the hyperactivity and impulsivity aspects of ADHD, had a more significant mediating effect on NSSI behavior through comorbid symptoms than ADHD_AD and ADHD_C. At the same time, among the three comorbid symptoms of AN, DD, and IAD, the mediating effect of AN on each ADHD subtype was the most obvious, followed by DD and IAD. All mediating effects had statistical significance after passing the Sobel test. These results indicate that ADHD treatment may alleviate NSSI behavior by reducing the severity of comorbid symptoms. Our research results are similar to those of co-disease research that found a relationship between N SSI and alcohol abuse/dependence (*Fulwiler et al., 1997*) and ADHD and alcohol abuse/dependence (*Biederman et al., 2010*). Other reports indicated that substance-addictive behavior (internet addiction) gradually germinated and worsened within the context of ADHD and finally led to the emergence of NSSI behavior. *Lam (2002)* also suggested that ADHD with substance abuse/dependence may be the link between self-injury and suicide. A study by *Izutsu et al. (2006)* showed that the prevalence of ADHD was higher among adolescents with self-injurious behaviors, and they had drug use in common. A longitudinal cohort study by *Hlund et al. (2020)* found that a large proportion of their cohort with both ADHD and NSSI behavioral characteristics also had varying degrees of substance dependence. Therefore, further understanding the potential mechanisms of comorbidity between ADHD and NSSI and alleviating their psychological and physiological comorbidity symptoms is crucial. In addition to substance dependence and addiction, psychiatric disorders with anxiety and depression as the main symptom play an important role in the relationship between ADHD and NSSI.

## Gender differences in NSSI mechanisms

In particular, the results of studies on the comorbidity mechanisms in patients of different sexes showed that the mediating effect of comorbidity was more obvious in girls, and females had significantly higher scores for various comorbid symptoms and ratios of the effects of various mediating pathways than males The sensitivity of the mediating effect of comorbidity symptoms in women also confirms the claim in the aforementioned studies that the incidence of NSSI is higher in the female population. Numerous studies showed that men and women regulate their emotions in different ways (*Augustine & Hemenover, 2009*; *Nolen-Hoeksema & Aldao, 2011*), which may contribute to the differential expression of psychopathology in men and women. For example, women's use of defensive self-focused thinking (*Nolen-Hoeksema, 2012*) may play a role in the higher incidence of depression

and anxiety in women than in men (*Aldao, Nolen-Hoeksema & Schweizer, 2010*; *Kessler, Merikangas & Wang, 2007*). Within the context of ADHD, more women choose to indirectly realize self-regulation of their symptoms through an accumulation of emotions. However, the accumulation of negative emotions tends to be accompanied by the generation and worsening of depression and anxiety and gradually leads to NSSI to achieve the release of negative emotions. *Hinshaw et al. (2012)* found that girls who exhibited significant clinical inattention had a significantly increased risk of developing NSSI at the end of adolescence compared to a normal sample. In the process of predicting all standard variables (namely NSSI, SI, and SA) through ADHD symptoms, *Meza, Owens & Hinshaw (2021)* found that the results were only significantly predictive of lifetime NSSI and SA. This finding shows that compared with the NSSI variable observed more frequently, ADHD symptoms will result in more serious and more direct forms of self-injury, especially in women (*Cho et al., 2008*). Therefore, whether it is the direct behavior of NSSI or the indirect comorbidity symptoms with ADHD, the female population seems to deserve more attention.

## Strengths & limitations

In the current study, the reasons for the influence of ADHD subtype symptoms on NSSI behavior were refined. Among patients with ADHD, subtypes with obvious attention deficit characteristics were more likely to exhibit NSSI behavior, whereas the hyperactive impulsive subtype had no direct impact on NSSI. The regression analysis results showing that BIS variables had no significant impact on NSSI also confirmed this conclusion. The symptoms of various subtypes of ADHD affect the occurrence of NSSI behavior through a variety of comorbid symptoms (depression, anxiety, and IAD), and ADHD treatment may achieve the goal of reducing NSSI behavior by reducing the degree of comorbid symptoms. Furthermore, young girls will be more likely to display NSSI behavior. These findings effectively reveal the complex interaction between ADHD and NSSI, as well as significant interaction differences between sexes , and provide effective theoretical support for the prevention and treatment of adolescent NSSI behavior. In addition, the results of this study were based on a relatively large number of participants recruited *via* random cluster sampling, which represents the characteristics of NSSI in middle school students in Zhejiang Province, China, and to some extent also represent other regions and cultures with NSSI problems. The current results can provide a reference for countries with similar cultural backgrounds.

However, there were several limitations to this study. First, this was a cross-sectional survey, and accordingly, it was not possible to draw conclusions regarding NSSI causes and effects, which requires longitudinal studies. Future studies may consider longitudinal prospective observations to address this limitation. NSSI is a heterogeneous disease that includes a wide range of behaviors with different clinical relevances. Although the measurement standard most used in studies clearly instructs participants to "please give a positive answer to the question only when you intentionally or intentionally hurt yourself," other measurement standards are not clear, and the purpose of the question is to identify intention self-harm. Therefore, these scales may not capture some atypical behaviors.

We hope that this research will inspire other researchers to develop new ideas and try to understand the diversity of sex difference effects in NSSI. Identifying these (multidimensional gender differences) will help promote new theories and may ultimately provide information for gender-specific prevention and intervention efforts aimed at reducing NSSI.

## CONCLUSION

In general, this work concluded: (1) teenagers' impulsivity may not be directly related to engagement in NSSI behavior. (2) Among the many subtypes of ADHD, attention deficit characteristics may be the main reason for increasing NSSI behavioral risk. (3) Each subtype of ADHD can affect individual NSSI behavior through comorbid symptoms, such as depression, anxiety, and internet addiction, and the mediating behavior is affected by sex.

In the current context, ADHD and its associated risk factors, such as anxiety, depression, and internet addiction, are the main factors that predict the outcome of self-injury before the patient's NSSI behavior causes serious personal injury. Identifying these salient features is a high priority of NSSI research and can play an important clinical role by guiding practitioners to identify patients with the highest risk and the greatest need for treatment. It is important to understand the risk process at the conceptual and theoretical levels and target NSSI high-risk adolescents at the clinical level, especially because high-risk adolescents do not usually seek professional help before or until NSSI causes serious personal injury (*Doyle, Treacy & Sheridan, 2015*). We must first pay attention to upstream prevention to increase the access of high-risk adolescents to care. First, an in-depth study of risk factors and their impact on NSSI is conducive to a better understanding and development of screening tools that can be used across the environment and assist healthcare practitioners in making effective diagnoses and providing treatment for complex conditions. Second, the priority is to understand the relevant mechanisms for maintaining NSSI to provide information for its prevention and treatment. Third, future research should give priority to risk factors such as females, those with anxiety, depression, and internet addiction, and the interaction between ADHD and risk and protective factors to provide the best understanding of and prediction of self-injury.

### Funding

This work was supported by the National Natural Science Foundation of China (NO. 81960262 and 82260878), as well as supports from International Scientific and Technological Cooperation Base for Precision Diagnosis and Treatment of Major Depression in Guizhou (QianKeHe [2018]5802), the High-level Innovative Talents Cultivation Program of Guizhou Province (QianKeHe [2016]5679), the Province Guiyang City Science and Technology Projects, the Zhu Subjects Contract ([2018]1-94), and the Guizhou Science and Technology Planning Project (QianKeHe [2020]4Y239). The funders

had no role in study design, data collection and analysis, decision to publish, or preparation of the manuscript.

## Grant Disclosures

The following grant information was disclosed by the authors:

National Natural Science Foundation of China: 81960262, 82260878.

International Scientific and Technological Cooperation Base for Precision Diagnosis and Treatment of Major Depression in Guizhou: QianKeHe [2018]5802.

High-level Innovative Talents Cultivation Program of Guizhou Province: QianKeHe [2016]5679.

Province Guiyang City Science and Technology Projects: [2018]1-94.

Guizhou Science and Technology Planning Project: QianKeHe [2020]4Y239.

## Competing Interests

The authors declare there are no competing interests.

## Author Contributions

- Fang Cheng conceived and designed the experiments, performed the experiments, analyzed the data, prepared figures and/or tables, and approved the final draft.
- Linwei Shi conceived and designed the experiments, performed the experiments, analyzed the data, prepared figures and/or tables, authored or reviewed drafts of the article, and approved the final draft.
- Huabing Xie performed the experiments, analyzed the data, authored or reviewed drafts of the article, and approved the final draft.
- Beini Wang performed the experiments, authored or reviewed drafts of the article, and approved the final draft.
- Changzhou Hu performed the experiments, authored or reviewed drafts of the article, and approved the final draft.
- Wenwu Zhang performed the experiments, authored or reviewed drafts of the article, and approved the final draft.
- Zhenyu Hu performed the experiments, prepared figures and/or tables, and approved the final draft.
- Haihang Yu performed the experiments, prepared figures and/or tables, authored or reviewed drafts of the article, and approved the final draft.
- Yiming Wang performed the experiments, prepared figures and/or tables, authored or reviewed drafts of the article, and approved the final draft.

## Human Ethics

The following information was supplied relating to ethical approvals (i.e., approving body and any reference numbers):

The Ethics Committee of Ningbo Kangning Hospital approval to carry out the study within its facilities (Ethical Application Ref: NBKNYY-2020-LC-52).

## Ethics

The following information was supplied relating to ethical approvals (i.e., approving body and any reference numbers):

The following information was supplied relating to ethical approvals (i.e., approving body and any reference numbers):

The Ningbo Kangning Hospital granted Ethical approval to carry out the study within its facilities (Ethical Application Ref. No.: nbknyy-2020-Lc-52, the year 2020).

## Data Availability

The codes and all the raw data are available in the Supplemental File.

## Supplemental Information

Supplemental information for this article can be found online at http://dx.doi.org/10.7717/peerj.16895#supplemental-information.

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
