# Peer review of "A study of the interactive mediating effect of ADHD and NSSI caused by co-disease mechanisms in males and females"

_PeerJ, doi:10.7717/peerj.16895_

## Round 0.1 · original submission · Minor Revisions

I have now received the reviewers' comments on your manuscript. They have suggested minor revisions to your manuscript. Therefore, I invite you to respond to the reviewers' comments and revise your manuscript.

Reviewer 1 ·

Basic reporting

-

Experimental design

-

Validity of the findings

-

Additional comments

I would like to thank you for inviting me to review this manuscript. I read the article carefully. Overall, I felt the research topic was interesting and the article appeared valuable. However, I am recommending that the paper is accepted for publication with minor revisions:

Before using acronyms, define them separately in the abstract and main text.

I believe that the use of the STROBE checklist might increase the quality of the submitted work and expedite the peer-review and publication process of this article.

Did authors perform power analysis? They should describe the sample size, power, and precision.

It is also unclear how your added research will dispel many of the misunderstandings surrounding this difficult situation since the data was only described and you did not contribute significantly to better manage these problems. Therefore, it is suggested that the discussion section be reconsidered in terms of essential items including practical Implications.

Moreover, there are minor concerns regarding English grammar and expression. So, I strongly recommend that the paper be checked by a fluent English speaker or professional English editing service before re-submission.

Reviewer 2 ·

Basic reporting

Dear author, accept my thanks. You have done a wonderful read. I enjoyed reading your article. I haven't read such an article in many years. Your attention to detail was excellent. Thank you again. I would suggest just a few things to improve your article:
1- Write the method and findings based on the STROBE statement.
2- State possible biases and explain how to manage them
3- Use logical scientific reasons in the discussion and help spread knowledge. Do not interpret findings based on existing knowledge alone.

Experimental design

Thank you again. I would suggest just a few things to improve your article:
1- Write the method and findings based on the STROBE statement.
2- State possible biases and explain how to manage them
3- Use logical scientific reasons in the discussion and help spread knowledge. Do not interpret findings based on existing knowledge alone.

Validity of the findings

Your attention to finding detail was excellent.

---

## Round 0.2 · accepted · Accept

In my opinion this manuscript has been revised with attention to the reviewers' comments and can now be published.

Reviewer 1 ·

Basic reporting

-

Experimental design

-

Validity of the findings

-

Additional comments

Many thanks for addressing all my comments.

Reviewer 2 ·

Basic reporting

I am glad to have had the opportunity to review your article. Requested corrections have been made. Congratulations to you.

Experimental design

Requested corrections have been made. Congratulations to you.

Validity of the findings

Requested corrections have been made. Congratulations to you.

Additional comments

Requested corrections have been made. Congratulations to you.